# Influence of Arc Brazing Parameters on Microstructure and Joint Properties of Electro-Galvanized Steel

**Soon Jae Lee [1] , Ashutosh Sharma [2] , Do Hyun Jung [1] and Jae Pil Jung [1],\***

1. Department of Materials Science and Engineering, University of Seoul, Seoul 02504, Korea; aolofinj@usc.edu.au (S.J.L.); jdh1016@uos.ac.kr (D.H.J.)
2. Department of Materials Science and Engineering and Department of Energy Systems Research, Ajou University, Suwon 16499, Korea; ashu.materials@gmail.com
* Correspondence: jpjung@uos.ac.kr; Tel.: +82-2-6490-2408

**Abstract:** Arc brazing of zinc-coated steel (EG, Electro-galvanized steel) using Cu-3 wt%Si filler metal was performed. The influence of arc current and brazing speed on the bonding properties of the joint, such as bead characteristics, arc penetration, joint hardness, and tensile shear strength were evaluated. The microstructural characteristics of the joint were examined by scanning electron microscopy (SEM) and the compositional information was revealed by energy dispersive spectroscopy (EDS). The throat thickness varies inversely with the brazing speed. The EG joint shows the formation of $Fe_2Si$ phases, which result in higher microhardness than the base metal. The tensile samples were fractured in base metal, while minor bead cracks developed in the samples brazed at 80 cm/min-80 A, 60 cm/min-70 A, 70 cm/min-70 A.

**Keywords:** gas metal arc brazing; zinc-coated steel; filler metal; tensile shear strength; microhardness

## 1. Introduction

The automotive industry always demands high reliability of its auto components to meet safety requirements. Various joints in automotive body parts, such as rooftops, hood, and car door frame are important parts requiring high reliability [1–6]. For joining steel sheets, fusion welding, friction, spot and laser welding have been used [7,8]. These joining methods are applied to automotive parts depending on their particular application and needs such as lightweight, high durability and corrosion resistance. However, these welding methods have several drawbacks, such as pore, spatter, flux residue, and corrosion after joining [7–10].

Gas metal arc (GMA) process is the most popular metal joining method using arc with a shielding gas due to its various advantages such as good weld penetration, reasonable brazing speed, less slag, etc., [11–13]. However, GMA brazing requires the optimization of various operating parameters, such as arc speed, arc current, filler feeding rate, shielding gas rate, etc. [13]. Among all these, heat input is regarded as the most important parameter that is related to arc current, voltage, and speed. For example, lower heat input results in a defect like insufficient weld penetration and unstable bead shape, while higher heat input degrades the base metals [14]. Therefore, the combination of arc current and brazing speed needs to be optimal for a promising brazed joint performance [14,15]. Meanwhile, zinc vaporization occurs from the Zn-coated steel (base metal) during arc welding processes because the melting temperature (460 °C) and vaporization temperature (906 °C) of Zn are lower than those of arc welding and arc brazing [16].

Therefore, in these welding processes, to control the heat input, cooling rate, and the joint gap is important to reduce zinc vaporization and defects [16–18]. Compared to previous studies, most of

the joining of Zn-coated steels were performed by using laser/arc or hybrid laser welding using Cu-Si filler. Our study reports a comprehensive study of arc brazing of advanced electro-galvanized steels (EG), its characteristics, properties and optimization of arc brazing current and speed for optimum joint reliability. A combination of brazing current and speed is required to obtain a strong and smooth brazed joint. Further, most of the studies are related to GI or GA steels, while studies related to EG steels brazing are limited in the literature. In this study, GMA brazing was performed on the Zn-coated steel sheets, and the brazing performance was studied through the investigation of bead appearance, microstructure, tensile shear test, throat thickness, and penetration width and depth.

## 2. Experimental Procedures

### 2.1. Materials

The base metal was an EG steel sheet (100 mm × 200 mm × 0.7 mm) provided by the POSCO Steel, Pohang, Korea. The thickness of the zinc-coated layer is approximately 3 μm. The filler used for the GMA process was CuSi-3 rod. The detailed composition of base metal and filler metal (Cu-3 wt %Si brazing filler) are listed in Table 1.

**Table 1.** The composition (in wt %) of the base metal and filler metal.

| Joining Components | Fe | Cu | Si | Mn | Pb | Zn | S | P | C |
|---|---|---|---|---|---|---|---|---|---|
| Base Metal | Bal. | - | 0.25 | 0.45 | - | - | 0.046 | 0.042 | 0.18 |
| Filler Metal | 0.03 | Bal. | 2.94 | 0.85 | 0.003 | 0.01 | - | - | - |

### 2.2. Arc Brazing Process

Figure 1 shows the GMA equipment (Daihen DP 350, OTC Daihen Asia Co., Ltd., Pathum Thani, Thailand). The various parts are shown to consist of a current-voltage controller, wire feeder and arc generator. The specimen was mounted on the brazing table and fixed by magnetic jigs. Argon (shielding gas) with 99.99% purity was blown at 20 L/min rate. The Ar gas and brazing filler wire were fed by the automatic system during the brazing process.

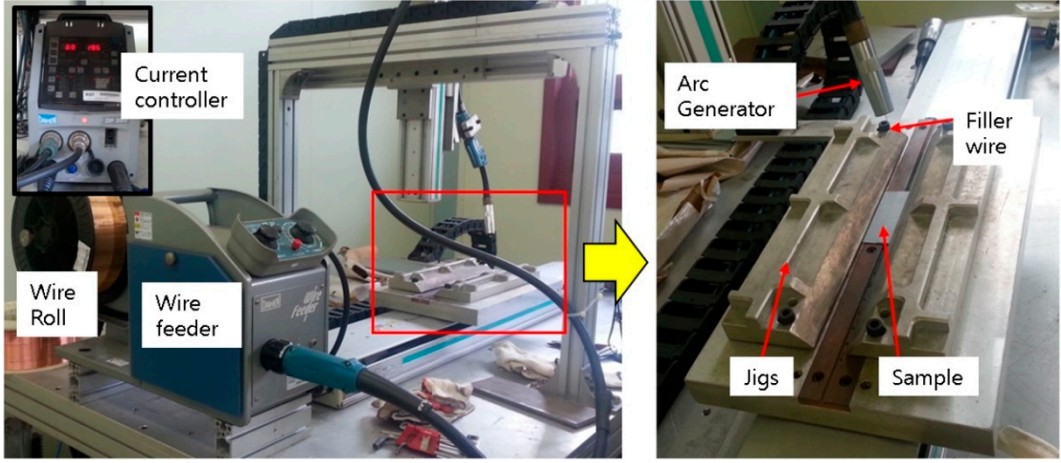

**Figure 1.** Arc brazing equipment and setup.

The arc voltage was 19.3–19.8 V and the length of the overlap joint in brazing specimen was 10 mm. There are two sets of variables studied for brazing, (1) arc current (50–80 A) and (2) arc travel

speed (brazing speed) of 60–90 cm/min, respectively. The heat input $Q$ (kJ/mm) can be calculated from the Equation (1) [19]:

$$Q = \frac{60 \times k \times V \times I}{1000 \times v} \tag{1}$$

where, $V$ and $I$ are arc voltage (V) and arc current (I); $v$ is the arc brazing speed (mm/min), and $k$ is the thermal efficiency = 0.9 (for GMA brazing).

### 2.3. Sample Preparation and Analysis

After the GMA brazing, the brazing specimens were cross-sectioned and epoxy mounted. The specimens were further ground with #500, #1200, #2400, #4000 sandpapers and polished with a diamond paste having 1 μm powder size. The specimens were etched with acid solutions of 10% $HNO_3$ and 10% HCl for 10 s. The microstructure of the cross-sectioned brazing joint was observed by scanning electron microscope (SEM, JEOL JSM-6480, JEOL, Tokyo, Japan), and its chemical composition was analyzed by energy dispersive spectroscopy (EDS, Oxford Instruments, Abingdon, UK). The penetration depth and width of the fusion zone was calculated and studied with various arc current and speed.

### 2.4. Mechanical Properties of Joint

#### 2.4.1. Microhardness

The microhardness was measured over 6 sections across the brazed joint depending upon the heat-affected zones (HAZ). The various sections, as shown in Figure 2, are parts of HAZ across the cross-section of the joint. Zone 1 is the filler portion, Zone 2 corresponds to the EG steel part near to the joint. Zone 3, Zone 4 and Zone 5 lie well ahead of the Zone 2. Zone 6 consists of the gap between the two-lap joints. XY shows the vertical line across joint cross-section for microhardness measurement. The hardness measurements were done using a hardness tester (Mitutoyo MVK H1) at a load of 100 gf and holding time of 15 s. The Vickers hardness is calculated as follows:

$$\text{Hardness (Hv)} = 1.854 \times \left(\frac{F}{d^2}\right) \tag{2}$$

where $F$ is the load (kgf) and $d$ is the average of the indentation diameter (mm).

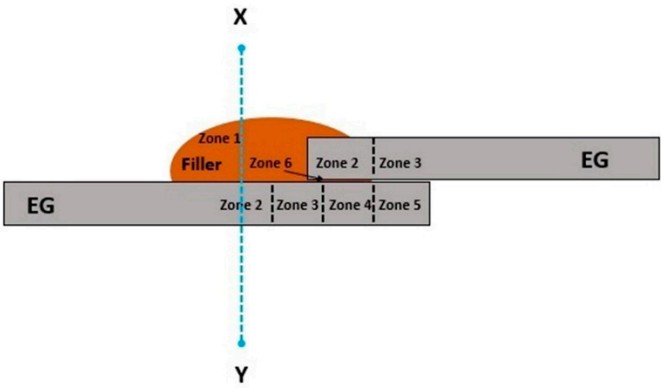

**Figure 2.** Microhardness measurement across the lap joint in various heat-affected zones (HAZs). XY shows the line profile for the hardness measurement.

#### 2.4.2. Tensile Shear Test

The tensile shear test was studied with a tensile testing machine (UTM, MTS 810, MTS, Eden Prairie, MN, USA) according to the Japanese standards reported in Refs [20–22]. The test samples were of 100 mm in length, 20 mm width, 0.7 mm thickness and 10 mm lap length. The strain rate was fixed to 0.001/s. The schematic of the tensile specimen used for shear is shown in Figure 3.

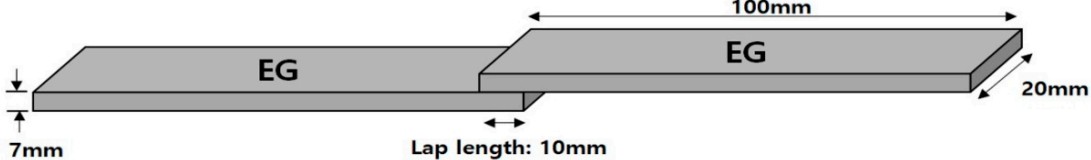

**Figure 3.** Sample dimensions for the tensile shear test.

## 3. Results and Discussions

### 3.1. Bead Characteristics

Figure 4a demonstrates the difference of bead surface with an arc current of 50, 60, 70 and 80 A at a fixed arc speed of 80 cm/min. Initially, the bead appearance was darker at 50–60 A. When the arc current reached 70 A, bead appearance was brighter with few instances of spattering. The bright color indicates smooth and clear joint surface. Further increases in arc current to 80 A shows even darker bead surface. Spattering occurs under many factors, such as an unstable arc, and/or unclean joint surface due to inadequate current/voltage supply, etc. [23,24]. The severe melting occurs at high current due to high input according to Equation (1).

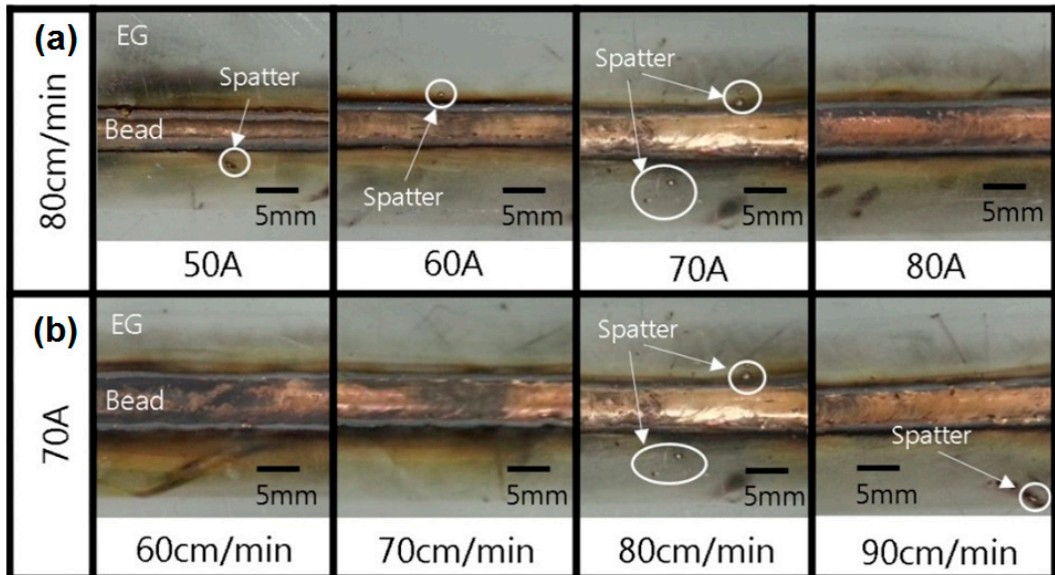

**Figure 4.** Bead appearance. (**a**) arc current (at 80 cm/min), and (**b**) brazing speed (at 70 A).

Schmidt et al. attributed this observation as a common concern in the welding of hot-dip galvanized steels [25]. Kimura and his co-workers reported that the reason for bead imperfections are due to zinc evaporation and propagation through the liquid filler metal [26].

Figure 4b shows the difference of bead face at various arc speeds for the constant arc current of 70 A. When the brazing speed was low, i.e., 60 to 70 cm/min, then the bead appeared dark covered with black residues. However, 80 cm/min of arc speed condition gave a bright and shiny color. In addition, faster brazing speed (90 cm/min) also produced dark beads. It is to be noted that heat input varies inversely with brazing speed. High heat input induces rapid melting and hence burnt out beads at low brazing speeds [14]. The optimum bead surface was obtained at 80 cm/min.

These findings of bead imperfections such as spattering and width irregularities have been verified by various researchers in the past but the detailed formation of bead characteristics is not available yet. Various types of zinc-coated steels play a role in laser welding but have not yet been researched for arc brazing in detail [27].

### 3.2. Bead Width

As already discussed, to stabilize the process monitoring and control of bead imperfections, we examined the bead width developed during the application of various brazing current and speed which controls the heat input and power. Figure 5a shows that the bead face width increases from 3.9 to 7.5 mm with the arc current from 50 to 70 A (at 80 cm/min). As already pointed out, this may be related to a high heat input which causes rapid fusion of base metal and Cu-Si filler producing thick Fe-Si intermetallic compounds (IMCs). Above 70 A, the bead width reduces slightly to 6.9 mm and remains stable.

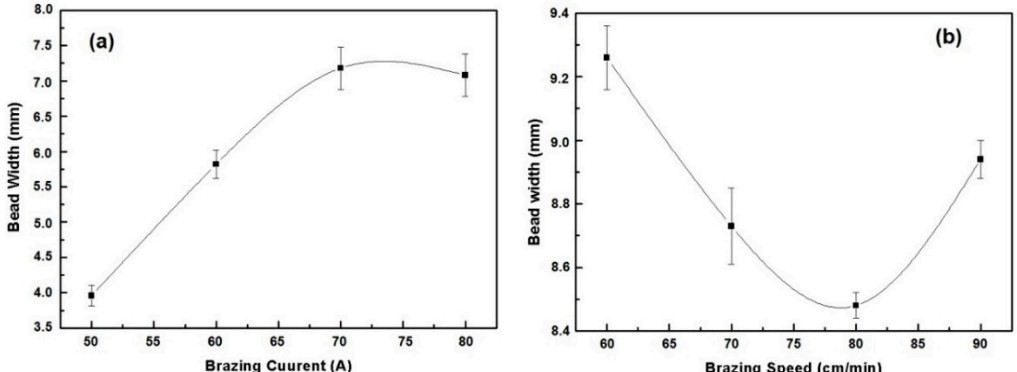

**Figure 5.** Bead face width as a function of (**a**) arc current (at 80cm/min), and (**b**) arc speed (at 70A).

Figure 5b shows that the bead face width reduces with brazing speed from 60 to 80 cm/min (arc current of 70 A). The bead width increases at 90 cm/min due to the turbulence experienced by arc flow at extremely high speeds. It is also noticed from Figure 5a,b that the difference of bead width at different arc current is approximately > 1 mm, and the smallest bead width is approximately ≈ 4 mm. On the other hand, the difference of bead width at different brazing speeds is << 1 mm and the smallest bead width is approximately 8.5 mm. It shows that arc current is a more important factor than brazing speed to control the bead width [14,28].

### 3.3. Penetration Depth and Width

Figure 6a shows the cross-sectional SEM image of bead showing fused zone with penetration width and depth at an arc current of 70 A/80 cm/min. The base metal is slightly damaged during GMA brazing due to the arc pool at high temperature. Therefore, proper arc current and speed are necessary even if the brazing temperature is over than the melting temperature of the base metal. The melting of base metal and filler creates a fusion zone, having a penetration depth and width.

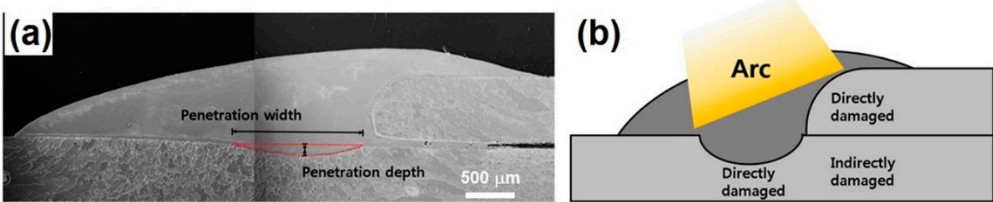

**Figure 6.** (**a**) Measurement of penetration width and depth, (**b**) Schematic showing damaged zones.

Figure 6b shows a schematic diagram to explain the damage to the base metals. Two types of damaged parts can be seen: the upper and lower side of the base metal, which are affected directly and indirectly by the arc. Normally, bead penetration occurs on the bottom side in a lap joint because of the HAZ with arc pool [17].

Figure 7a shows the penetration depth and width increase with arc current except at 70 A due to progressive fusion of Cu-Si filler with the EG steel. The penetration depth and width are maximum at 80 A. Lesser penetration indicates less amount of heat energy and vice versa.

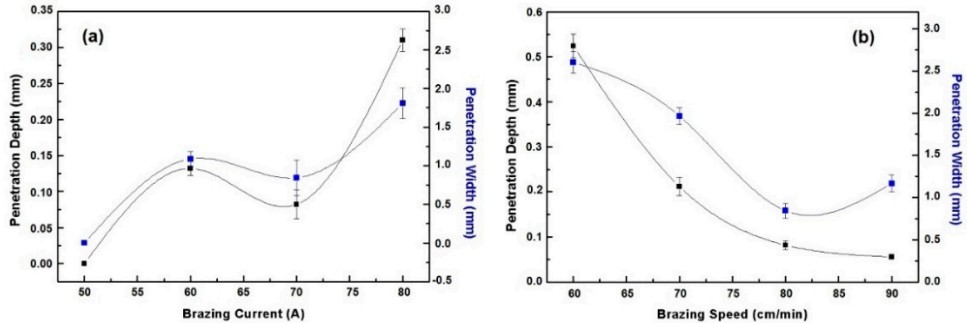

**Figure 7.** The variation of penetration depth and width with (**a**) arc current (at 80 cm/min) and (**b**) speed (at 70 A).

Figure 7b shows a significant decrease in the penetration depth and width, especially penetration depth decrease is more than width. As discussed, the penetration is related to the heat input amount according to the Equation (1). The brazing speed varies inversely to the heat input. The heat input is tremendous at lower brazing speeds, and overheating causes severe penetration. Base metal's thermal diffusivity is the most important for penetration sensitivity because of cooling rate. Further decrease in penetration depth at 90 cm/min with an increase in width is caused by a lesser fusion of filler with base metals and unstable arc flow.

Thermal diffusivity ($D$ in m$^2$/s) is proportional to base metal characteristics, such as thermal conductivity ($k$) and inversely proportional to thermal capacity ($c$) according to Equation (3).

$$D = \frac{k}{\rho c} \tag{3}$$

where, $k$ = thermal conductivity (W/(m·K)), $\rho$ = density (kg/m$^3$) and $c$ = specific heat capacity (J/(kg·K)). According to Equation (3), the penetration value can be controlled by base metal selection. Due to the high vapor pressure of the zinc-coated on steels, plasma formation may occur and affects the absorption of arc energy. The maximum vapor pressure can rise up to 120 MPa around ≈2600 °C. The viscosity of iron at such a high temperature is given by relation:

$$\eta = \eta_0 \, e^{\frac{E}{RT}} \tag{4}$$

Here, R is the gas constant, $E$ is the temperature coefficient of viscosity, and $\eta_0$ is the initial viscosity. According to equation (4), $\eta \approx 4 \times 10^{-3}$ kg·m/s. As a result, the spattering and porosity rise but penetration reduces and bead discontinuity increases at high brazing speeds [14].

### 3.4. Microstructural Analysis of Joint

The microstructure of bead joint was examined at optimal penetration width and depth observed at arc brazing condition of 70 A-80 cm/min. Figure 8a–d shows the microstructure of the joint, fusion zone and the various Fe-Si IMCs identified by the EDS analysis (Figure 8e–h). The Gibbs free energy of various Fe-Si reaction compounds is shown in Table 2.

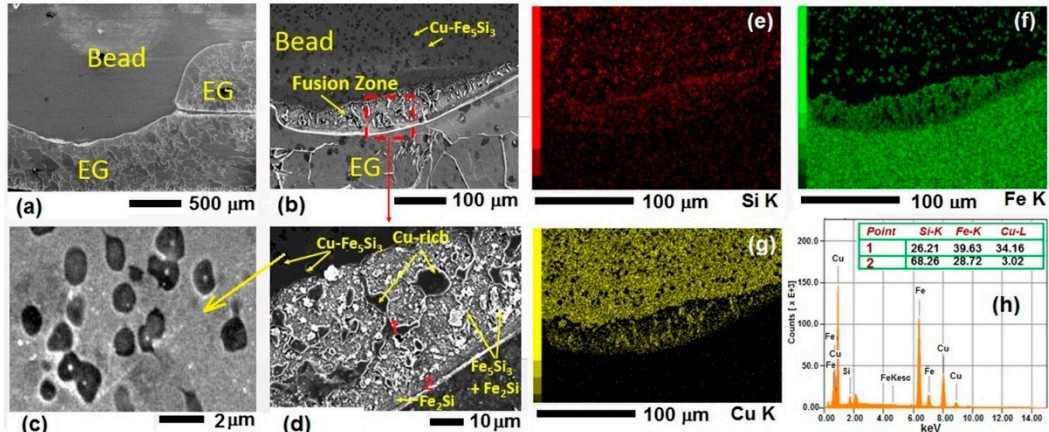

**Figure 8.** The microstructure of the bead cross-section and its composition. (**a**) The SEM image of lap joint, (**b**) the fusion zone, (**c**) the morphology of flower structure CuFe$_5$Si$_3$, (**d**) high-resolution image of fusion zone. (**e**–**g**) elemental maps of Si, Fe, and Cu obtained from SEM image of (**b**), (**h**) the EDS spectrum of (**d**). The atomic percent values are given in the table.

**Table 2.** Gibbs free energy of various Fe-Si compounds [14].

| S.No. | Reaction | $\Delta G$ (kJ/mol) |
|---|---|---|
| 1 | Fe$_2$Si + Si → Fe$_5$Si$_3$ | −285.2 |
| 2 | Fe$_2$Si + 3Si → 2FeSi$_2$ | −29.42 |
| 3 | Fe$_5$Si$_3$ + 7Si → 5FeSi$_2$ | −30.6 |
| 4 | Fe$_5$Si$_3$ + Fe → 3Fe$_2$Si | −104.68 |

According to Sharma et al. there are various possibilities of Fe-Si compounds according to the Gibbs thermodynamics between Fe-Si reactions, such as the Fe$_5$Si$_3$ and Fe$_2$Si and the Fe$_5$Si$_3$ (Reactions 1–4, Table 2). Out of these, Fe$_5$Si$_3$ is the primary phase in Zn-coated steel as expected using Cu-Si based filler [14]. It is a noteworthy point to mention that Cu is randomly present surrounding FeSi3 phases via liquid phase separation around ≈1100 °C. Therefore, the final phase designated inside the weld is Fe5Si3(Cu) [29–32].

The filler metal area contains scattered flower petals like particles (Fe$_5$Si$_3$) [30,31]. In addition, at the base metal side (EG) has high Fe concentration and low Si concentration. The filler metal side includes Fe compound (Figure 9e–h) while the base metal side does not contain Cu. Fusion zone is the copper matrix reinforced with Fe$_5$Si$_3$ generates at higher arc current and speed, i.e., (80 A-80 cm/min). This may be attributed to the higher current and speed, the melting and solidification are rapid, and arc and liquid metal flow are turbulent. The EDS mapping analysis shows the atomic ratios identified in different regions are approximately ~5:3 and ~1:2 as shown in Figure 8 h. This confirms the presence of two Fe-Si compounds, Fe$_5$Si$_3$, and Fe$_2$Si as expected in a matrix of Cu. Therefore, rapid segregation of Fe$_5$Si$_3$Cu particles is established in the bead zone and grows there. These segregated compounds grow into spherical form by an Ostwald ripening mechanism to reduce their surface energy.

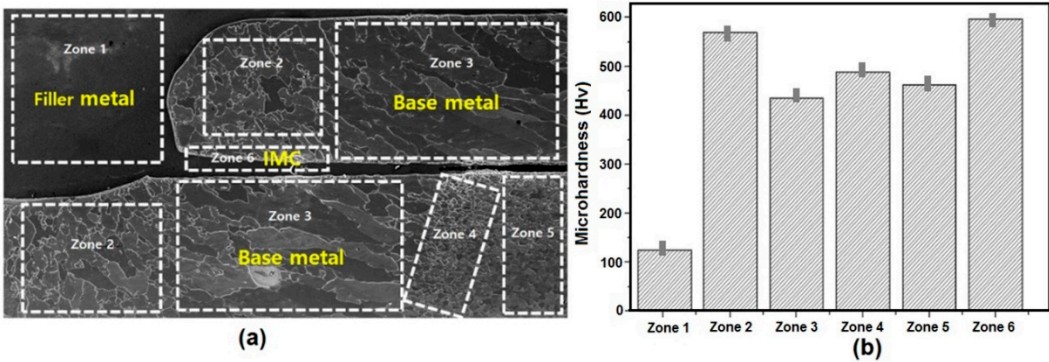

**Figure 9.** (**a**) The section for microhardness measurements. (**b**) Microhardness of the cross-section
(80 cm/min, 70 A).

### 3.5. Joint Mechanical Properties

#### 3.5.1. Microhardness

Figure 9a shows the cross-section of the various zones in the bead and corresponding hardness
values are plotted in Figure 9b. Zone 1 represents the filler metal area and has the lowest microhardness
around 122.6 Hv. The microhardness values of Zone 2 and 6 are almost same and maximum (around
596.6 Hv). These Zones 6 and 2 are under direct contact to the laser beam and fused with the filler
metal. Zone 6 has the maximum hardness of 596.6 Hv due to the reinforcement of $Fe_5Si_3$ with Cu of
the filler metal. This can be also related to the finer grains brought about by the composite effect of
$CuFe_5Si_3$ compounds. Other Zone 3, 4, and 5 have microhardness varying from 435.4 to 488.2 Hv
(Figure 9b). The hardness of Zone 3 is slightly higher than Zone 2 and 6. This is already clear from the
bigger grains as shown in Figure 9a. Zone 4 and 5 have similar microhardness as they are away from
the fusion zone.

The various zones are the parts of HAZ across the cross-section line of the joint. As shown in
Figure 9a, the recrystallized grains are present in the base metal. The microstructure of the base metal
consists of a mixed coarse and fine grains. The coarse grains show the process of ongoing grain growth
during the joining. While the filler metal shows relatively smaller grains than that in the base metal.

Zone 1 is the filler itself while Zone 2 is the EG steel portion near the joint. Zone 3, Zone 4 and
Zone 5 lie in the same line away from the joint (Zone 2). Zone 6 is the gap between the two EG steel
laps. As shown in Figure 9a, Zone 2 has the fine grains while Zone 3 has a large-grained structure, the
grain size of Zone 3 is bigger than that of Zone 2. In addition, Zone 4, and 5 have the smallest grain
size because of the typical HAZs. Generally, the smaller grain has a harder microhardness value. The
microhardness of Zone 2 and 6 are highest because of the additional hardening contribution coming
from the reinforcement of hard $Fe_5Si_3$ particles into the base metal during fusion. In addition, the more
Si component is contained in Zone 2 because it is close to the filler metal. It means Zone 2 has more
Fe-Si compound than Zone 4. According to these results, Fe-Si compound is related to microhardness
increase, and it is more influential than grain size effect towards hardness [14].

#### 3.5.2. Joint Tensile Shear

Figure 10a displays the post-test photographs tensile shear fractured samples. All the samples
after tensile shear testing were fractured at the base metal. The bead thickness (few mm) is too thick
compared to the base metal thickness (≈0.7 mm). Tensile shear strength results indicate the base metal
strength. Some cracks are also found in the beads (80 cm/min-80 A, 60 cm/min-70 A, 70 cm/min-70 A).
Figure 10b depicts the tensile shear testing results of the different investigated samples.

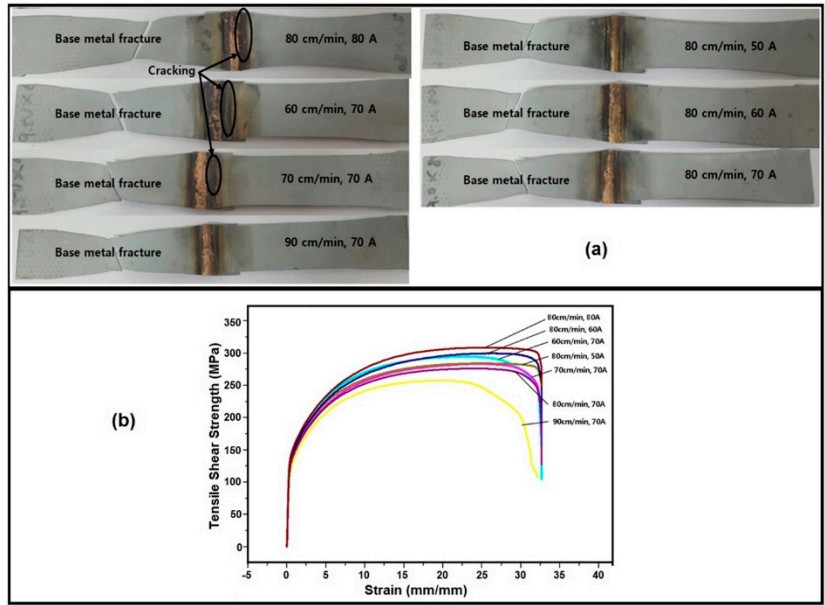

**Figure 10.** (**a**) Post-test photographs of the tensile samples, and (**b**) Tensile shear stress-strain curves of the various joints.

The UTS of the samples varies from (from 240 to 300 MPa), and a minute increment in elongation. Lowest UTS is obtained at 70 A/90 cm/min followed by 70 A/80 cm/min due to a lesser time available for the arc at high temperature. The UTS further decreases at lower arc current from 50 A/80 cm/min and increases slightly for lower brazing speeds (70 A/60 cm/min).

It can be noticed that the arc current and the brazing speed are interrelated to each other as arc current is directly proportional to the heat input while brazing speed is related inversely to the heat input. As the arc current increases and the brazing speed declines, the joint strength improves due to sufficient heat input. Tensile strength is maximum ($\approx$320 MPa) at brazing speeds of 80 A/80 cm/min followed by 60 A/80 cm/min due to a maximum penetration of base metal. This result is consistent with the results reported by Li and his co-workers for welding of galvanized steel sheets [31]. As shown in Figure 10a, base metal fracture occurs in all the samples. This signifies a higher joint strength over the base metal (EG) caused by the formation of $Fe_5Si_3$ IMCs.

### 3.5.3. Strengthening Mechanism of Weld Joints

The effective contribution to the various strengthening mechanisms operating in the weld joints can be expressed as:

$$\sigma = \sigma_{Hall\text{-}Petch} + \sigma_{dislocations} + \sigma_{precipitates} + \sigma_{solid\ solution}. \tag{5}$$

where the various terms represent yield strength due to (A) Hall-Petch (grain boundary), (B) dislocations, (C) precipitates (Orowan), and (D) solid solution strengthening, respectively.

The various contributions (A–D) to the yield strength are given by:

$$(A): \sigma_{Hall\text{-}Petch} = \sigma_0 + kd^{-1/2} \tag{6}$$

where $d$ is the grain size, and $\sigma_0$ is a constant related to the hindrance to motion of dislocations. Hall-Petch coefficient, $k = G \cdot b\theta$. $G$ = 78 and 66 GPa for EG and fusion zone. Burger vector for fcc metals, $b$ = 0.254 nm [33,34]. The misorientation angle $\theta$ = 3° and 15° for the base metal, and the fusion zone, respectively. The Hall-Petch contribution is given by:

$$\sigma_{Hall\text{-}Petch}\ (EG\ base\ metal) = 59.4d^{-1/2} \tag{7}$$

$$\sigma_{Hall\text{-}Petch}\ (fusion\ zone) = 25.1d^{-1/2} \tag{8}$$

$$(B):\ \sigma_{dislocations} = M\alpha Gb\rho/^{1/2} \tag{9}$$

Substituting $M \approx 3$ and $\alpha = 0.3$ for fcc metal, $b = 0.254$ nm, and $\rho$ the dislocation density $\approx 1.57 \times 10^{15}$ m$^{-2}$ [35].

$$\sigma_{dislocations}\ (fusion\ zone) = 59.1\ \text{MPa}. \tag{10}$$

$$\sigma_{dislocations}\ (base\ metal) = 70.8\ \text{MPa}. \tag{11}$$

$$(C):\ \sigma_{precipitates} = M\frac{0.4Gb}{\pi\sqrt{1-v}}\frac{\ln(2\langle r\rangle/b)}{\lambda} \tag{12}$$

where Poisson's ratio of steel matrix, $v = 0.29$ [36], and $<r>$ is the average radius of a circle for a spherical precipitate [37]:

$$<r>^2 = 2/3\ r^2 \tag{13}$$

and $\lambda$ inter precipitate spacing [38]:

$$\lambda = 2<r>\ (\ (\pi/4f)^{1/2} - 1) \tag{14}$$

As there was no precipitation in EG base, the Orowan bypassing contribution is:

$$\sigma_{\text{precipitate}}\ (\text{fusion zone}) = 48.3\ \text{MPa and (D): } \sigma_{solid\ solution} = AG\varepsilon^{4/3}{\cdot}C^{2/3}\ [35] \tag{15}$$

where $A$ is of the order of 0.1, $C$ is the solute concentration, $C = 0.45$, $G = 78000$ MPa, $\varepsilon$ is the lattice strain to overcome the size difference (Mn (atom radii = 161 pm) and Fe (atom radii = 156 pm) ~3%. Based on these variables, the $\sigma_{solid\ solution}$ of the fusion zone and base metal are 36.7 and 41.4 MPa. Therefore, according to equation (4), the total strength is equal to 303 MPa, which is higher than the base metal strength ($\approx$240 MPa) and hence base metal fracture is obtained in the samples. The predicted strength of EG lap joints shows a good correlation with the experimental data.

## 4. Conclusions

Heat input controlled by arc current and speed and current results in increased penetration of the base metal. Increase in arc current (50 to 70 A) or a decrease in arc speed (70 to 60 cm/min) causes more fusion of EG steel and Cu-Si filler. The cross-section of the EG joint displays the formation of $Fe_5Si_3$ and $Fe_2Si$ intermetallic compounds. $Fe_2Si$ compounds exist on the base metal side. Microhardness of the joint is enhanced by the reinforced Fe-Si intermetallic particles into fusion zone and grain refining by the arc heating effect. In addition, microhardness is approximately > 90% hardness than that of the EG steel. Tensile shear tests indicate higher shear strength at the interface of the base metals (EG) and $CuSi_3$ filler metal zone. The tensile fractured samples show a base metal fracture irrespective of brazing condition except minor cracks seen near edge of the bead brazed at 80 cm/min-80 A, 60 cm/min-70 A, and 70 cm/min-70 A. The predicted strengthening mechanisms operated in the tensile welds support the experimental findings.

**Author Contributions:** J.P.J. provided the concept and facility for carrying out this work. S.J.L. performed the experiments, methodology and prepared original draft of this manuscript. A.S. analyzed the data and provided the mechanism of the brazed joint formation. D.H.J. performed formal analysis and editing of the manuscript. J.P.J. reviewed and recommended the final corrections. Finally, all authors read, checked and corrected the whole manuscript.

**Funding:** 2018 Research Fund of the University of Seoul.

**Acknowledgments:** This work was supported by the 2018 Research Fund of the University of Seoul.

**Conflicts of Interest:** The authors declare no conflict of interest.

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
