# Peer review of "Influence of Arc Brazing Parameters on Microstructure and Joint Properties of Electro-Galvanized Steel"

_metals, doi:10.3390/met9091006_

Round 1

Reviewer 1 Report

Thank you for the opportunity to review this manuscript. The authors have performed some interesting experimental work and provide some meaningful results. Please consider the following points toward strengthening the manuscript.

All equations in the text are poorly formatted. For instance, the authors should italicize the variables as is the common convention. Overall the English of the text is good, but some minor editing is required. The experimental procedure and the results are of very good quality. Certainly the manuscript provides meaningful data toward process optimization with weld quality and weld properties in mind. However, can the authors provide more in-depth correlation between the process parameters and fundamental material behavior? For example, the Aims and Scope of Metals is to provide “a forum for publishing papers which advance the in-depth understanding of the relationship between the structure, the properties or the functions of all kinds of metals.” In its current form, the manuscript provides worthwhile observations, but does not necessarily provide the in-depth analysis that Metals seeks.

Author Response

Reviewer report 1

Thank you for the opportunity to review this manuscript. The authors have performed some interesting experimental work and provide some meaningful results. Please consider the following points toward strengthening the manuscript.

Answer: The authors would like to thank the reviewer for this comment.

All equations in the text are poorly formatted. For instance, the authors should italicize the variables as is the common convention.

Answer: The equations are now formatted properly as per recommendation.

Overall the English of the text is good, but some minor editing is required.

Answer: The English of the text is now improved.

The experimental procedure and the results are of very good quality. Certainly the manuscript provides meaningful data toward process optimization with weld quality and weld properties in mind. However, can the authors provide more in-depth correlation between the process parameters and fundamental material behavior? For example, the Aims and Scope of Metals is to provide “a forum for publishing papers which advance the in-depth understanding of the relationship between the structure, the properties or the functions of all kinds of metals.” In its current form, the manuscript provides worthwhile observations, but does not necessarily provide the in-depth analysis that Metals seeks.

Answer: Thanks for this comment and recommendation. The authors have now improved the discussion for various structure related properties.   

Reviewer 2 Report

Arc brazing was performed by authors that a basic understanding on the microstructures and properties of brazed steel joints was achieved. Given its potential implications in automotive industry, it is a interesting research. I would suggest this article to be accepted after some minor revisions:

1, Yes, typos were still remained such as page 1 line 16: should be "Fe2Si phases, which"? oage 2 line 56, CuSi-3 or Cu-3Si? page 4 line 130: "bead face width increases rises from 3.9"-what the rises stand for here? I believe the author can eliminate those during revision process.

2, Figures are not clear. I dnot know the reason but the quality of figures should be improved. In particular, Fig.8 (b) (c) and (d) were taken at same location in different magnification? if so, may combine them. Also, a SEM image that clearly shows the location and morphology of Fe-Si IMCs will be needed because the mechanical properties of brazed joints ar closely related the precipitation of "flowerlike shaped particles"?  No error bar was given in the mechanical properties results? 

3, Page 7 line 197 "...driven to central bead region creating porosity inside the bead matrix." I cannot see the porosity in the draft. If author put this in discussion, better to make it more clear?

4. A more comprehensive understanding on the relationship between microstructures and mechanical properties should be presented. In section 3.5,  the mechanical properties were listed merely, it shoud be closely discussed in regards with the microstructure result, i.e., what the micristructural similarity between present work to the reference 31 that was on aluminum?

Author Response

Reviewer report 2

Arc brazing was performed by authors that a basic understanding on the microstructures and properties of brazed steel joints was achieved. Given its potential implications in automotive industry, it is a interesting research. I would suggest this article to be accepted after some minor revisions.

Answer: The authors would like to thanks the reviewer for this comment.

1, Yes, typos were still remained such as page 1 line 16: should be "Fe2Si phases, which"? oage 2 line 56, CuSi-3 or Cu-3Si? page 4 line 130: "bead face width increases rises from 3.9"-what the rises stand for here? I believe the author can eliminate those during revision process.

Answer: This has been corrected now. It is Cu-3wt%Si filler. Commercially available and represented in the form CuSi-3 rod.

2, Figures are not clear. I do not know the reason but the quality of figures should be improved. In particular, Fig.8 (b) (c) and (d) were taken at same location in different magnification? if so, may combine them. Also, a SEM image that clearly shows the location and morphology of Fe-Si IMCs will be needed because the mechanical properties of brazed joints are closely related the precipitation of "flowerlike shaped particles"? No error bar was given in the mechanical properties results?

Answer: Figures quality have been increased now. Fig. 8 (b) (c) and (d) are at same magnification and merged now. The figures have been re-arranged and improved. The error bars have been added now. The SEM image of scattered flower petals particles 8 (c) have been added.

3, Page 7 line 197 "...driven to central bead region creating porosity inside the bead matrix." I cannot see the porosity in the draft. If author put this in discussion, better to make it more clear?

Answer: Thanks for this comment. The authors have deleted this sentence to avoid confusion.

A more comprehensive understanding on the relationship between microstructures and mechanical properties should be presented. In section 3.5, the mechanical properties were listed merely, it should be closely discussed in regards with the microstructure result, i.e., what the micristructural similarity between present work to the reference 31 that was on aluminum?

Answer: The authors have added more discussion now in the revised manuscript. The reference 31 has been changed now with more suitable for steel joints.

Round 2

Reviewer 1 Report

I believe that the authors have added discussion and analysis that have improved the manuscript and have demonstrated the relationship between microstructure and properties. There are some minor formatting issues throughout the paper, such subscripting relative to variable references and equations, that need to be addressed.

Author Response

Review Report

Comments and Suggestions for Authors

I believe that the authors have added discussion and analysis that have improved the manuscript and have demonstrated the relationship between microstructure and properties. There are some minor formatting issues throughout the paper, such subscripting relative to variable references and equations that need to be addressed.
Answer: The authors would like to thank the reviewer for this recommendation. The authors have checked the manuscript thoroughly and corrected the subscripting errors. Some equations have been reformatted again.
